# Electronic Communication with Public Administration in the Time of COVID-19—Poland’s Experience

**DOI:** 10.3390/ijerph18020685

**Published:** 2021-01-14

**Authors:** Aleksandra Klich

**Affiliations:** Faculty of Law and Administration, University of Szczecin, 70-240 Szczecin, Poland; aleksandra.klich@usz.edu.pl

**Keywords:** electronic communication, serving documents, public authority, administrative proceedings

## Abstract

The situation associated with the growing number of Severe acute respiratory syndrome coronavirus 2 (SARS-CoV-2) infections forced ongoing monitoring of the epidemic situation, which entailed an introduction of a number of restrictions and solutions intended to isolate the infected persons on the one hand, and to minimize the risk of development of an epidemic in Poland on the other. Activity of the Polish legislator is also essential, which tried to introduce solutions that would correspond with current expectations and needs. Given the multiplicity of the introduced regulations, interpretation of provisions of statutes has not always been easy. In this paper, the author points to the issues of communication with a public authority by specific reflections on the principles of serving documents on beneficiaries of EU programs under which they were awarded funding for their implementation on the basis of EU regulations addressing the use of the European Regional Development Fund, the European Social Fund and the Cohesion Fund for programs implemented as part of the cohesion policy (Regulations of the European Parliament and of the Council (EU) of 17 December 2013: no. 1303/2013, no. 1301/2013, no. 1304/2013, no 1300/2013, and no. 1299/2013). The author focuses on the issues of communication with an authority in a situation where administrative proceedings are initiated against a beneficiary of EU funds, e.g., for returning the granted funding. The author points to the dynamics of the legislator’s work in this respect by analyzing the rules for serving documents by a public authority on beneficiaries who are public entities and those who are not. The author’s main research aim is to analyze existing provisions establishing the possibility of electronic communication with a public authority, and also to assess them critically due to the extraordinary situation caused by Coronavirus Disease 2019 (COVID-19). This is intended to verify the main research hypothesis focusing on the attempt to answer a question whether existing regulations, and those created at the time of the epidemic threat and the state of epidemic in Poland facilitate citizens’ electronic communication with a public authority. The author aims to answer a question about whether the Polish legislator responds appropriately to the numerous emerging challenges associated with the pandemic and whether it created regulations that effectively ensure the possibility of continuity of contact with a public authority for citizens who are the beneficiaries of public funds. This analysis may contribute to the understanding of whether and how it is possible to improve citizens’ contact with public authorities, which in the future may eliminate barriers and obstacles arising in this regard. The author bases her reflections on the experience resulting from providing legal services for one of the Polish Managing Authorities of the Regional Operational Programme using at the same time a number of research methods (i.e., the method of interpretation of applicable laws to establish applicable provisions of the law that regulate admissibility of electronic communication with a public authority and to establish efficiency of such communication, the analytical method, applied in reference to the relevant state of the art in the achievements of legal scholarship, and the empirical method, based on observation and analysis of practical issues resulting from the author’s cooperation with a Polish managing authority). In her conclusions, the author points to the lack of introduction of comprehensive regulations (also at the EU level—for all EU Member States) in terms of de-formalizing the principles of communication in the course of pending administrative proceedings. The author notices an absence of unambiguous regulations that allow for a scanned document signed by hand and sent my email to be qualified into the category of documents served by electronic means, through use of means of electronic communication. The author assesses this absence negatively due to the fact that such action seems the simplest in a situation caused by COVID-19.

## 1. Introductory Issues and Establishing Research Problems

The situation associated with an epidemic threat caused by Severe acute respiratory syndrome coronavirus 2 (SARS-CoV-2) has affected almost all spheres of every-day functioning due to its global character. This also applies to the operation of public authorities, which are in this time obliged under the provisions of applicable law (including the Constitution of Poland) to act with regard to matters (not only administrative) entrusted in them, including maintaining continuity [1]. The threat concerning the development of the epidemic escalated in Poland in the first half of March 2020. On 14 March 2020, the state of epidemic threat was introduced in Poland applicable until 20 March 2020 when the state of epidemic was introduced in the territory of the Republic of Poland. A growing number of infections, as well as deaths, triggered anxiety manifested, i.e., in the Polish government’s taking a decision on specific isolation of society, which also involved laying down numerous rules that intended to ensure safety and to minimize the possibility of the spread of infections. Undoubtedly, the coronavirus pandemic (COVID-19) is a serious challenge for all societies. In order to stop the spread of the virus, governments of many countries have adopted policies intended to regulate human behavior and their social habits. In particular, citizens throughout the world are strongly encouraged to get involved in the so-called “social distancing” [2]. This term is also known in international literature as “physical distancing” [3,4,5]. It is indisputable that activities that aim to create new unknown rules of functioning in society were not sufficient. As a consequence, it became necessary for the legislator to respond to the transformations on an on-ongoing basis, which was to prevent a state of affairs in which regulations would not keep up with the reality. This means that relevant regulations which mitigate the difficulties emerging recently in many areas of everyday life must be introduced. A lack of an adequate strategy and activity of the Polish legislator may lead to a paralysis of operation of public authorities or programs guaranteeing funding from EU resources that the beneficiaries are implementing. Responding to COVID-19 is an unprecedented challenge for public sector practitioners and addressing those challenges requires knowledge about the problems public sector workers face [6]. The legislator’s activity in this period was hugely intensified and unfortunately it was not always met with coherent and well thought-through solutions, which on the one hand is justified by the extraordinary nature of the situation in which Polish society found itself, and on the other, it highlighted the flaws and imperfections of temporary solutions, thus leading to a multiplicity of interpretation doubts after revoking regulations created ad hoc (e.g., in terms of suspending deadlines for both legal and procedural acts). However, first one needs to emphasize that under the Regulation of the Minister of Health of 13 March 2020 on announcing the state of epidemic threat on the territory of the Republic of Poland, the state of epidemic threat was announced throughout the country on 14 March 2020 due to SARS-CoV-2 infections. In turn, by the Regulation of the Minster of Health of 20 March 2020 on announcing the state of epidemic on the territory of the Republic of Poland, the state of epidemic throughout the country was announced until further notice. On the same day, by the Regulation of the Minister of Health of the same date, the state of a pandemic threat was announced which has not been cancelled to date.

The analysis led to stablishing that this study relates to the situation of beneficiaries, i.e., stakeholders receiving aid by obtaining resources from funds ensuring support under the European Union cohesion policy and aims to answer the following basic research questions:Has COVID-19 had a negative impact of the contact of EU programs’ beneficiaries with public authorities and did it force implementation of relevant regulations that facilitate communication by electronic means?Is the beneficiary’s legal form (i.e., is it a natural person or a local government unit) relevant to the choice of the form of communication with a public authority?How can the beneficiary communicate with a public authority in a situation where physical contact at an office is impossible? Can a public authority force the electronic form of communication on the beneficiary?What factors can contribute to improving communication with a public authority? What is essential in creating regulations that support electronic communication in public administration?Does sending a scanned signed document determine the meeting of requirements of electronic service and may it determine effective electronic communication?

In order to answer these questions, it is necessary to conduct am appropriately directed analysis based on juxtaposing and comparing regulations in force before introducing the state of epidemic threat and epidemic in Poland with regulations created as a result the pandemic caused by COVID-19. The main source of the research material involves the author’s experience resulting from providing legal services for one of the Managing Authorities in Poland which involved requests for legal opinions addressed to the author by the authority’s employees and practical problems the author noticed as a result of practicing the profession of a legal counsel. Observation of the occurring transformations and of the fossilization of legal regulations led to a number of conclusions, including a primary one about the legal provisions not being adjusted to the dynamic situation associated with COVID-19. The dynamic situation associated with the pandemic also caused an absence of scholarly studies in which authors would analyze the situation of beneficiaries of e.g., EU funds against whom proceedings for returning those funds were initiated during the pandemic. At the moment there are no studies that analyze the problem of the situation of beneficiaries against whom proceedings are pending, e.g., for returning the obtained funds and those who are requesting relevant funding for a planned project. Regardless of the reason substantiating contact with a public authority, due to COVID-19 and the implemented mechanisms and personal protection measures, these entities do not have a possibility of direct contact, which may potentially lead to exclusion, e.g., from a competition announced in a specific activity or it may lead to the stakeholders’ real deprivation of the possibility to defend their rights due to the pending proceedings. For this reasons, capturing the comprehensive array of possibilities of contact with public administration by means of modern forms of communication is crucial. In this scope, the ongoing communication with an authority which proceeds by means of the “SL2014” system (Central teleinformation system), the main application of the central technical solutions, including information and communications technology ICT system used, i.e., in the process of clearing the project and communicating with the Regional Operational Program’s Managing Authority, must be distinguished from communication caused by the initiation of administrative proceedings, e.g., for returning the awarded funds.

The issue that has triggered most doubts since the first days after the announcement of the state of epidemic threat in Poland and then the state of epidemic included the way in which beneficiaries can submit electronic documents to the authority in order for this action to be effective in administrative proceedings conducted by the managing authority. This must refer to communication with public and non-public entities alike. To answer the question about how a beneficiary of EU funds may communicate with a public authority in a situation where personal contact in the office was reduced to close to zero, it is necessary to establish to what degree implementation of solutions based on modern technology is possible. For this reason, the analysis of existing regulations aims to answer the question about whether in pending administrative proceedings in which that far the authority had not served documents to the public electronic registry box (called ePUAP) but had been doing so in a traditional manner (through the Polish postal operator), the beneficiary should be informed (and how) on the change of the manner of serving documents (which now were to be served with the use of ePUAP). With regard to public entities and the public authority’s communication with them, it is also important that in principle documents may be served on such an entity solely to its electronic registry box—without the need to obtain its consent for serving documents in such a way. This was possible before the announcement of the state of epidemic in Poland and was not only introduced due to the coronavirus outbreak. Moreover, such a public entity cannot resign from submitting documents by means of electronic communication. For this reason, as regards communication with public entities, the issue that needs to be settled is whether the managing authority may oblige the party initiating the proceedings (e.g., in the event of filing an application for a case to be re-examined or for granting a relief) in a traditional form to submit documents electronically. The next area of doubts concerns authenticating documents in the beneficiary’s communication with a public authority, that is both the possibility to send correspondence by email in the form of a scanned document which has been signed by hand (if the entity does not have a qualified electronic signature) upon meeting one of the requirements enumerated in Article 39^1^ § 1 of the Polish Code of Administrative Procedure (hereinafter also: the Code, [7]), (Dz. U. (Journal of Laws) of 2020 item 256), and in the matter of signing documents by the authority by using a certified electronic signature where the beneficiary does not have the possibility to verify its authenticity because they do not have relevant software. The said doubts serve as an example. It is because they emerged in the initial period of introducing restrictions resulting from the announcement of the state of epidemic threat and the epidemic in Poland. Their source was the managing authority’s striving to encourage beneficiaries to submit correspondence through the ePUAP platform or to contact the authority by means of electronic mail, as well as to enhance the possibility of using such communication channels in pending administrative proceedings. Due to absence of guidelines in this area these doubts intensified in the first days after the announcement of the state of epidemic threat and then epidemic, which is extremely crucial given the broad scale of using EU funds.

## 2. Impact of COVID-19 on Applicable Regulations.

Legislative changes first forced by the state of epidemic threat and as a consequence the state of epidemic covered a wide range of issues. As the virus spread, representatives of governments throughout the world introduced significant restrictions, i.e., on the movement of people, the functioning of services and rules on physical distancing or application of personal protection measures. The importance of services which use modern technological solutions increased in the public space. In the current reality, technology has a profound effect on citizens’ daily lives and ensures their access to information and communication i.e., with competent authorities [8]. It needs to be emphasized, that research on the use of information technology in order to improve the efficiency of and enhancing trust in public administration was the subject of analysis long before the development of the pandemic caused by COVID-19, which is testimony to the significance of the so-called informatization of the public services sector too [1,9,10]. Among the key areas which feature numerous legislative changes one must identify most of all the judiciary, the entrepreneurs market, local government, and the public procurement sector. In the practice of the so-called professional attorneys—in-fact representing their clients, the most essential changes which have affected the to-date model of operation concerned the manner of organization and conducting court proceedings, both before administrative courts and common courts of law. These changes have also covered the manner of operation of public authorities. Given the possibility of obtaining financing under European funds, the basic aim of this analysis aiming to provide answers to the problematic questions is to point to the rules and regulations of conducting administrative proceedings in cases which deal with the implementation of programs under the cohesion policy financed in the 2014–2020 perspective. From the point of view of the research hypothesis the scope of freedom of a public authority in contact with entities participating in the implementation of these programs, which was moved to virtual space, is especially important. As an introduction it is worth emphasizing that in accordance with applicable laws, the coordination of implementation of operational programs in Poland is the responsibility of the minister competent for regional development who performs the tasks of a Member State. In turn, their implementation, including preparation of draft development strategies of a voivodship (regional level of local government) and of other development strategies, regional operation programs, programs that serve the implementation of partnership agreements for a cohesion policy, and their execution lies with the governing body of a given voivodship (called the managing authority of the regional operational program—hereinafter: the managing authority). A voivodship is a unit of administrative division of the highest level in Poland, since 1990 - a unit of basic territorial division of administration, since 1999—also a local government unit. Persons seeking financial support for projects carried out under a given Project Priority Axis, described in detail in an application for funding for project implementation, found themselves in a rather negative position due to the epidemic situation. The implementation of provisions included in a project financing agreement made with a governing body of a given voivodship was often problematic, which is naturally associated with obstacles that have their source both in terms of moving on to subsequent stages of project implementation (e.g., due to an absence of real possibilities of undertaking activities aiming to file a final application for payment) and in pending administrative proceedings, e.g., for returning awarded funds.

Undoubtedly, conducing administrative proceedings in the time of a coronavirus (COVID-19) epidemic causes a lot of problems which all public authorities who apply the provisions of the Code of administrative procedure and their employees must face. This statement aims to answer affirmatively the first research question concerning the impact of COVID-19 on the situation of beneficiaries of EU funds due to the need to communicate with a public authority. This results primarily from the fact that a lot of organs (and the offices that serve them) are not only tackling staff shortages (caused either by sickness or the need to look after their children while schools, nurseries, and other educational institutions are closed) but also from the need to limit, if not completely then at least to the minimum, contact in the authority (authority employee)—party (parties) to administrative proceedings relation [11]. An analysis concerning the first hypothesis relating to the issue is possible when taking into account practical problems concerning the rules of organization and work of public authorities in Poland in the time of a state of epidemic threat and the state of epidemic caused by COVID-19. The basis of reflections in this area includes the most important regulations of the following acts:a)Act of 2 March 2020 on special solutions related to preventing, counteracting, and combating COVID-19, other infectious diseases and emergencies caused by them (Dz. U. (Journal of Laws) of Laws) of 2020 item 374 as amended, hereinafter: Shield),b)Act of 31 March 2020 on amending the act on special solutions related to the prevention, counteracting and combating of COVID-19, other infectious diseases and crisis situations caused by them, as well as some other acts (Dz. U. (Journal of Laws) of 2020 item 568 as amended, hereinafter: Shield2),c)Act of 16 April 2020 on special support instruments in connection with the spread of the SARS-CoV-2 virus (Dz. U. (Journal of Laws) of 2020 item 695, hereinafter: Shield3),d)Act of 14 May 2020 amending some acts in the field of protective activities in connection with the spread of the SARS-CoV-2 virus, (Dz. U. (Journal of Laws) of 2020 item 875, hereinafter: Shield4).

## 3. The Authority’s Electronic Communication With Parties to the Proceedings

The requirements for serving documents electronically were formulated by the legislator in Article 39^1^ of the Polish Code of Administrative Procedure before the announcement of the state of epidemic threat and epidemic. Pursuant to Article 39 ^1^(1) of the Code, service of documents shall be effected by means of electronic communication if a party to proceedings or other participant in the proceedings shall satisfy one of the following conditions:a)he submits the application in the form of an electronic document via the electronic registry box of the public administration authority;b)he applies to the public administration authority for such service and provides the authority with his email address;c)he consents for documents in the proceedings to be served by such means and provides the authority with his email address.

The Polish legislator understands means of electronic communication as technical solutions, including information and communications technology (called ICT) and compatible software tools, which allow for individual remote communication with the use of data transmission between ICT systems, in particular electronic mail. The public administration authority may request that a party to or other participant in the proceedings expresses his consent for documents to be served in the electronic form prescribed in other categories of individual matters specified and disposed of by this authority. A public authority may also request that such consent be given be sending such a request by means of electronic communication to the e-mail address of the party to or other participant in the proceedings. When analyzing if the beneficiary’s legal form (i.e., is it a natural person or a local government unit) is relevant to the choice of the form of communication with a public authority, it is worth emphasizing that the legislator conditions the effectiveness of service on confirmation of receipt of the electronic document, that is in compliance with the instruction from the public administration authority on the manner of receiving documents, especially the manner of identification at the indicated electronic address in the public administration authority’s ICT system and according to the information about the requirement to sign the official confirmation of receipt as specified. Admittedly, effectiveness of serving pleadings in the form of electronic documents is conditioned on confirming receipt of such pleadings in a manner specified by the legislator. Nevertheless, the possibility for pleadings to be served in the form of electronic documents to the electronic address of the addressee is conditioned on sending a notice to the electronic address of the addressee containing: (a) an indication that the addressee may receive the document in electronic form; (b) an indication of an electronic address which may be used by the addressee to download the document and confirm its receipt; (c) an instruction in relation to the manner of receipt of the document, in particular the manner of identification at the indicated electronic address in the ICT system of the public administration authority and information concerning the requirement to sign the official confirmation of receipt as specified.

The requirement of signing an official confirmation of receipt “as specified” means the requirement of placing a qualified electronic signature or a signature confirmed be the ePUAP trusted profile (with regard to a qualified electronic signature). If the document has not been received in electronic form as specified by statute, the public administration authority shall, after seven days of the day when the notice was sent, send another notice about the possibility to collect the document, where it needs to be borne in mind that such a notice may be automatically created and sent by the authority’s ICT system and receipt of this notice is not confirmed. If the document has not been received, service shall be deemed effective upon the expiry of 14 days of the date when the first notice was sent. If a document in the electronic form prescribed has been deemed received, the public administration authority is obliged to allow the addressee to access the following in the ICT system of the authority:a)the electronic document for a period of at least three months of the date when the electronic document was deemed collected;b)information regarding the date when the document was deemed collected;c)information on dates when notices were sent.

If a party to the proceedings or another participant in the proceedings is a public entity (including local government units) which is obligated to provide and maintain an electronic registry box service shall be made via the electronic registry box of such entity and the provisions of the afore-mentioned Article 39^1^ of the Polish Code of Administrative Procedure do not apply. This leads to an unambiguous answer to the question posed at the beginning of this fragment according to which the legislator conditions the choice of the beneficiary’s communication with a public authority on whether it is a public entity. In such cases it has an absolute obligation to communicate by means of an electronic registry box. However, when juxtaposing legal provisions with practice, it needs to be concluded with great sadness that public entities do not respect this obligation, which will be discussed in further parts of this study. These solutions, as has been pointed out, had been formulated before the announcement of the state of epidemic threat and the state of epidemic. Therefore, they are not specific facilitations introduced by the legislator due to the occurrence of COVID-19. We need to note that, e.g., the North-West Regional Development Agency in Romania delegated attributions through the framework agreement with the Ministry of Public Works, Development and Administration (Managing Authority); during the state of emergency an electronic network was created between beneficiaries—the North-West Regional Development Agency and the Managing Authority, for each of the stages related to the evaluation, implementation and monitoring of the Regional Operational Program’s projects. At the beginning of June, a new module for the implementation of the Regional Operational Program’s 2014–2020 projects was added to the MySMIS online platform, making it possible for beneficiaries to submit documents in electronic format, without having to bring written originals to the Agency’s headquarters (requests for reimbursement/payment/pre-financing), thus contributing to the complete digitization related to the implementation of the EU-funded projects (https://www.interregeurope.eu/improve/news/news-article/9327/digital-jump-in-romania-due-to-covid-19/; access on 30.12.2020 r.). Such simplifications were not fundamentally introduced in Poland because the “SL2014” application was being used, which—as has been pointed out—is employed i.e., in the process of clearing the project. Whereas when it comes to the implementation of individual solutions that facilitate communication in the process of, e.g., recovering funds and the need to contact the managing authority, no detailed solutions were introduced either. It is because the ePUAP is in place and in operation and electronic communication is possible, which simply got revived in the time of COVID-19. It did not become a norm, though. The author tried to obtain information from Polish Managing Authorities so as to attempt to evaluate the scale of communication by electronic means, though she did not obtain statistical information, which is due to the lack of ongoing monitoring of the problem.

## 4. Rules for Serving Documents on Non-Public Entities

Continuing reflections focused on the research problem addressing the conditioning of the choice of communication with a public authority on whether the beneficiary is a public entity or not, first we need to note that the legislator, in the aforementioned Article 39^1^ § 1 of the Polish Code of Administrative Procedure, specified in a very restrictive and firm way the principles and possibilities of public administration authorities’ using the measure of serving documents by means of electronic communication. In consequence, if the premises of Article 39^1^ § 1–1a of the said Code are met, all documents in administrative proceedings which are addressed to any participant of these proceedings can be served by using means of electronic communication, which includes technical solutions, including ICT devices and their corresponding software tools, which allow individual remote communication by using data transmission between ICT systems and in particular electronic mail. The Polish legislator directly points out that in the case of participants to proceedings other than a public entity serving documents by means of electronic communication is dependent on the consent of the addressee of the authority’s document (that is participant to proceedings). This consent may be revealed in three ways, which is by

a)clear action expressed in submitting a request in the form of an electronic document via the electronic registry box of the public authority (hereinafter: authority),b)requesting at the authority for such service along with providing the authority with an electronic address, andc)expressing consent for documents to be served by means of electronic communication along with providing the authority with their electronic address.

In consequence we may assume that in the light of regulations applicable on the ground of the Polish legal order, serving documents in administrative proceedings by the authority by means of electronic communication in the first period of the state of epidemic threat and epidemic fundamentally was possible when two conditions were met; that is, the participant in the proceedings have given his consent and the authority is equipped with an ICT system that ensures operation of service of electronic documents in the scope that allows identification of system users (i.e., addressees of documents) by using a trusted signature, information verified by means of a qualified certificate of an electronic signature or other technologies applied in the system used by the authority. The legislator expressly distinguishes between two situations, since the participant’s acceptance may—apart form a clear submission of a request in the form of an electronic document—take the form of him requesting at the authority or him expressing his consent for the authority. The difference between these two forms involves the fact that in case of a request the initiative lies with the proceedings’ participant, whereas “expressing content” happens as a result of a summon addressed to him by the authority which conducts the proceedings [12]. A provision formulated like this excludes the possibility of “coercing” the proceedings’ participant to use the possibility of serving documents by means of electronic communication. A reverse situation would be inadmissible, that is if the authority served a decision in the form of an electronic document despite the fact that the party had not requested this and consent for it had not been obtained. Such a service would be ineffective because one could not acknowledge that the party had the opportunity to learn its content.

If service of documents by means of electronic communication is to occur on the initiative of the authority, it is necessary to ask the participant to give his consent for such a solution. The subject of the authority’s request may involve a question about consent to have documents served be means of electronic communication in a certain case or other categories of individual cases, specified by the authority, dealt with by the authority. The second case involves obtaining consent for this form of service also in connection with other cases which may be conducted in the future. The wording of Article 39^1^ of the Polish Code of Administrative Procedure does not exclude the possibility of the authority requesting the participant to consent to documents in these proceedings to be served by means of electronic communication at all stages of proceedings. As a rule, the authority’s request for the consent should have a written form and be served in a manner specific for this. The legislator provided for a possibility to apply for the consent by means of electronic communication to the email address of the party to or other participant in the proceedings. This is possible when the authority knows the email address of a given person. However, what is significant is that principles of effective service of an electronic document laid down by the legislator do not apply in such a situation. The legislator directly excludes application of these rules, which means that the authority’s mere application for the consent does not have to have the form of an electronic document yet. This solution surely encourages simplification and acceleration of the procedure of sending the request to the participant and in consequence it should allow for a quicker response from the participant in terms of providing answers. This simplification concerns only the authority’s service of its request on the participant but not the submission of a document with the participant’s declaration to the authority.

It seems that the legislator does not introduce a time limit on applying to the participant of the proceedings to express his consent for service of documents by means of electronic communication. This means, that de facto such an application may be sent at any stage of administrative proceedings. This is confirmed by the content of Article 63 § 5 subsection 1 of the Polish Code of Administrative Procedure, which is a certain guarantee that allows the proceedings’ participant to make a suitable choice of the manner of communication with the authority at each stage of the proceedings. The addressee of the authority’s application has the possibility to give his consent or refuse to give his consent in the analyzed subject matter. The consent must be expressed in a way that does not raise doubts and must be clear. However, it needs to be emphasized that failure to take a stand by the addressee of the authority’s application will be equal in legal consequences with the absence of the necessary consent. It is not possible to accept the construct of the so-called tacit consent. Therefore, if the participant of the proceedings does not give his express consent which does not raise doubts, then any presumptions by the authority about expression of consent for service by means of electronic communication is inadmissible. Due to the fact that the authority’s application is a type of a summon, provisions of Article 54 of the Code will apply to it, where elements of summons have been specified. Doubts may only be raised by the question of specifying the time limit within which the request should be complied with. However, taking into account the fact that the consent for electronic service may be given at any stage of pending proceedings, it is impossible to restrict the right to submitting such a declaration with a time limit. As has been pointed out, where the summoned participant to the proceedings remains silent, as well as in the event of an express refusal, it will be reasonable to serve a pleading (i.e., document in a paper form) in a manner and according to rules laid down in the act.

It is also important that the authority is obliged to instruct the participant about what shall be understood as means of electronic communication in order not to give him a wrong impression that communication, e.g., by sending an email to the authority with scanned documents attached will be effective. In the light of applicable laws one cannot assume that sending a scan of a signed document means that rules on electronic service have been met. A scan must be identified with an electronic copy of a paper document, whereas applicable laws do not provide for the possibility to authenticate such a copy by a public authority. Without a doubt, when a scanned document is sent through, in principle it will not have a secure electronic signature or a signature placed by hand, which is contrary to electronic service discussed in Article 39^1^ of the Code. The analysis concerning the matter in question leads to a hypothesis according to which the authority which applied to the party to the proceedings in order to obtain consent for electronic service should inform the party clearly and in detail about what the possibility of using electronic service entails. What is essential here is the provisions which in fact do not apply to the request itself sent by means of electronic communication to the electronic address of the party to or other participant in proceedings in order to obtain consent for serving documents in the form of electronic documents, but which are a determinant for deeming electronic service effective. For this reason, the authority should in particular inform the party about the electronic address from which the addressee may download the document and under which he should confirm service of the document, the manner of receipt of the document and in particular the manner of identification at the indicated electronic address in the ICT system of the public administration authority and information concerning the requirement to sign the official confirmation of receipt as specified.

The principle of the written form specified in administrative law is also essential for this analysis because the wording of applicable laws clearly shows that the authority’s correspondence with parties to administrative proceedings does not have to proceed as traditional written correspondence. The legislator stipulates for a possibility to handle administrative matters not only in a traditional written form or in an electronic form which is its equivalent value, but also in other less formalized ways. Matters may be disposed of orally, by phone, by means of electronic communication, or by other means of communication, if it is in the interest of the party and no provision of law provides otherwise. The contents and essential reasons for such disposal shall be entered in the records by way of minutes or annotation signed by the party. Legal scholars and commentators express a view according to which taking into account the currently announced state of epidemic it is worth considering using them in some cases, especially in minor administrative matters, e.g., by sending rulings to the email address of the party upon obtaining their consent for such a way of communication and with regard to provisions of Article 46 § 5 and 6 of the Code. [1].

When summing up the analysis intended to establish what factors may contribute to improving communication with a public authority and what is important in crating regulations that support electronic communication in public administration, it is worth emphasizing that the legislator did not stipulate directly in the initial weeks of announcing the state of epidemic threat and the state of epidemic in Poland the possibility of deeming a scanned document sent by a party which is not a public entity as effective service. No regulations aiming to deformalize contact through bringing it down to communication using electronic mail were introduced either. For this reason, taking into account the wording of Article 7a and Article 81a of the Polish Code of Administrative Procedure, it is reasonable to settle any doubts to the benefit of the party of administrative proceedings. Observation of the authority’s actions in this scope makes it possible to state that most of the authorities in Poland have acted and are acting so indeed. Legitimization of such action is also included in Article 7a of the Code, in which the legislator introduced a specific principle of friendly interpretation of regulations in order to reduce the risk of burdening the party with negative effects of unclear provisions.

## 5. Rules for Serving Documents Electronically on Public Entities

The legislator introduced in the content of the norm of Article 39^2^ of the Polish Code of Administrative Procedure a general obligation to use the measure of electronic service with the use of an electronic registry box, which applies when the party to or other participant in the proceedings is a public entity obliged to make available and operate an electronic registry box. For this reason, where the party of administrative proceedings is a public entity, it is obliged to make available and operate an electronic registry box. The legislator used an enumeration in this provision which clearly specifies entities classified in the category of public entities on the basis of the criterion of performing public tasks. Serving electronic documents on these entities will proceed on the basis of premises specified in Article 39^1^ of the Code, laid down for addressees of documents who are not public entities.

The wording of Article 39^2^ of the Code may determine the need to serve documents on public entities to their electronic registry box’s address. It is possible to conclude that this regulation is lex specialis vis-a-vis principles of service specified in Article 39 of the Code. However, one needs to refer here to the stipulated principle which shows that the legislator adopted two equal ways for the authority to serve its documents, i.e., serving documents in the written form and serving an electronic document by means of electronic communication. Each of these ways results in effective service of a document, i.e., service which allows the addressee to learn the content of the document in the proceedings [13]. The wording of Article 39^2^ of the Code provokes formulation of a thesis according to which it is the authority which conducts the proceedings that has the right to choose how to serve documents on a public entity, which especially substantiates the postulate of implementing the principle of quick and simple proceedings. The said principle demonstrates that the authority, having different possibilities of action, first chooses those which implement this principle specified in Article 12 of the Code to a greater degree [12]. Undoubtedly, electronic communication best and most effectively allows implementation of the principle of quick and simple proceedings. This is why if the authority, for factual or legal reasons, deems it more reasonable to serve documents in a traditional manner, Article 39^2^ of the Code will not be an obstacle there. It needs to be emphasized that the wording of this regulation, regardless of whether service to the electronic registry box is treated obligatorily or optionally, does not facilitate questioning the validity of serving a document in the traditional form if its addressee has learnt the content of the document served in such a way. Regulations concerning electronic service do not exclude the principle of the written form understood as a possibility to have the matter dealt with also in a form other than an electronic document [14,15].

Where previous correspondence with a public entity has been done in the traditional form, i.e., service to the electronic registry box has not been employed, these actions naturally remain in force and are effective. However, it needs to be postulated, especially given the drafted amendments which will be addressed in the last part of the study, that public entities should carry out contact with the authority using the electronic registry box. For this reason, in my opinion, the authority’s contact with public entities should involve transferring all communication in this regard to electronic circulation. Effectiveness of service to the electronic registry box of a public entity will be established in accordance with the rules applicable to serving documents be means of electronic communication. Reflections on the moment in which electronic communication with stakeholders who are not public entities can be initiated remain valid in this regard. It seems that also in this case (that is where a public entity obliged to use an electronic registry box is a party to proceedings), the legislator does not introduce a time limit on applying to the participant for his consent to have documents served by means of electronic communication. This means, that de facto such an application may be sent at any stage of administrative proceedings. This is confirmed by the content of Article 63 § 5 subsection 1 of the Code, which is a certain guarantee that allows the proceedings’ participant to make a suitable choice of the manner of communication with the authority at each stage of the proceedings.

## 6. Serving a Print-Out of a Document

According to Article 14 of Shield2, Article 39^3^ was added after Article 39^2^ in the Code of Administrative Procedure, which is a manifestation of the legislator’s response to practical problems in the beneficiaries’ communication with public authorities. The aim of the regulation is to make it possible for public authorities to deal with all matters in the form of electronic documents using the ICT system even if conditions determining the possibility of applying rules concerning electronic service applicable before the state of pandemic are not met. From 18 April 2020, i.e., from the day of applicability of this provision, each document issued by the authority in the course of administrative proceedings may be in the form of an electronic document. However, it is also essential that the manner of serving this document will have one form when the party to or other participant in the proceedings satisfies the following requirements: submits the application in the form of an electronic document via the electronic registry box of the public administration authority; applies to the public administration authority for such service and provides the authority with his email address or expresses consents for documents in the proceedings to be served by such means and provides the authority with his email address (in such a case a document is served by means of electronic communication), and it will have a different form if such premises are not met or where the party or other participant has resigned from serving documents by means of electronic communication [15]. The legislator constitutes in the newly added Article 39^3^ of the Code a solution unknown thus far to the Polish legislation involving a manner of serving documents issued in the course of the proceedings in the form of an electronic document different to service by means of electronic communication. It is service in a traditional way (i.e., on the basis of Article 39 of the Code, according to which a public authority serves documents by registered post via a postal operator) of a print-out of a document issued in the form of an electronic document. Such actions confirm the hypothesis presented in the course of the analysis, according to which the Polish legislator, seeing problems in communication of beneficiaries of EU programs with public authorities, created new methods that improve this communication. This activity of the Polish legislator deserves credit because it takes into account potential digital exclusion of parties to the proceedings, which do not have the technical facilities or practical skills that allow them to read a document issued in an electronic form.

In accordance with Article 39^3^ § 2 of the Code, service of a print-out of a document is possible in the case of documents issued by a public authority in the form of an electronic document using an ICT system, which has a qualified electronic signature, trusted signature or a personal signature, an advanced electronic seal or a qualified seal appended to it. In such situations service may involve serving a print-out of a document obtained from this system which reflects the content of this document. However, this is on the condition that the party or other participant to the proceedings have not submitted an application in the form of an electronic document via the electronic registry box of the public administration authority, they have not requested at the public administration authority that such service be made and have not given their consent for serving documents in such a way. The document’s obligatory elements include: (a) information that the document was issued in the form of an electronic document and was signed by an electronic signature, a trusted signature, or a personal signature, providing the first and second name and the post of a person who signed it, or that the document was appended with an advanced electronic seal or a qualified electronic seal; (b) document’s ID assigned by the ICT system through which the document was issued. Optionally, the print-out of the document may include a mechanically recreated signature of the person who signed it.

As a consequence, the legislator legalized the authority’s serving a print-out of a document obtained from the system that reflects the content of the document issued by the authority in the form of an electronic document. However, it is essential for the print-out to come from the ICT system with the use of which the authority issued the document in the form of an electronic document and must reflect the content of this document. The content of the print-out must be identical to the content of the electronic document [15]. The possibility of serving print-outs concerns situations in which both the party, by its action, and the authority did not strive to be able to implement electronic service under Article 39^1^ of the Code. Despite the fact that the legislator refers to premises that allow electronic service to entities that are not public, this regulation should not be treated in a narrowing way. It seems reasonable to state that the discussed regulation will apply also in the case of service to public entities, though in these cases the rule should be serving to the electronic registry box, pursuant to Article 39^2^ of the Code. In any case, a print-out of a document should be proof to what has been stated in the document issued in the electronic form.

## 7. Conclusions

Technology is crucial for the protection of the community, which involves the fact that digital tools must ensure protection of citizens and must serve to level out social and economic divisions and to promote necessary transformations that aim to implement the set objectives. Moreover, local and regional authorities, when creating new legislative solutions, must act in a way that prevents potential digital exclusion, as a result of which unequal treatment of citizens might take place. In order to bring local administration and public services closer to the citizens, and also in order to facilitate communication in all directions, it became necessary to accelerate the process of digitalization as part of internal and external structures and processes [7]. Globalization, the growth of international competition, technological and information changes undoubtedly cause the transformation of forms and concepts of society. In consequence, public administration is becoming flexible, decentralized, market-based, and democratic [16]. It also needs to be noticed that at the EU level no comprehensive regulations in terms of de-formalizing the principles of communication in the course of already pending administrative proceedings have been introduced.

Referring to the subject-matter of the analysis and the situation of the beneficiaries who are not public entities, first and foremost one needs to point to the lack of possibility of “pushing them” to use electronic service. In order for service referred to in Article 39^1^ of the Polish Code of Administrative Procedure to be possible, it is necessary that the beneficiary should act clearly by submitting a request in the form of an electronic document or that he should give his consent for such a solution. The second requirement concerns in particular situations where serving documents by means of electronic communication is to proceed on the initiative of the authority.

The authority writing to the beneficiary under Article 39^1^ of the Code should, as has been demonstrated, instruct him on the consequences of expressing consent to electronic service (including also in terms of the need to have appropriate software). Where the beneficiary gives his consent to electronic service, the burden of directing and receiving documents to and from the authority lies on him. It also needs to be remembered that failure to take a stand by the addressee of the authority’s application will be equal in legal consequences with the absence of the necessary consent. It is not possible to accept the construct of the so-called tacit consent. Therefore, if the participant of the proceedings does not give his express consent which does not raise doubts, then any presumptions by the authority about expression of consent for service by means of electronic communication is inadmissible.

When analyzing the matter in question, one needs to see a distinct absence of unambiguous regulations that allow qualification of a scanned document signed by hand and sent my email in the category of a document served by electronic means, using means of electronic communication referred to in Article 39^1^ of the Code. Taking into account the wording of Article 7a of the Code or 81a of the Code, documentation aiming to eliminate the possibility of abolishing the presumption of effective service should be produced extremely carefully. One needs to remember that not each beneficiary uses Outlook, which makes it impossible to assume unequivocally that the function of read receipt will be sufficient to assume efficiency of service. Naturally, as much as functionality of a given electronic mail allows such a formula, it is an additional aspect which determines minimization of the risk of abolishing the presumption of effective service. One needs to approach this form of communication with great caution due to the administrative court’s potential questioning of such a formula of serving the authority’s correspondence. It seems that in such a case a system of serving documentation in a way that eliminates the likelihood of leaving given post “without a response” (e.g., should a given employee of the authority be absent) should be created. For this reason, it is recommended that solutions guaranteeing and facilitating effective communication with the authority, regardless of unexpected situations or absence of an employee of the authority, be drafted.

Undoubtedly, the state of epidemic threat and the applicable state of epidemic enforce introduction of solutions that affect negatively the correctness of pending administrative proceedings as little as possible, which is evidenced by, for instance, the wording of Article 39^3^ of the Code or by promotion of the so-called hybrid service. For this reason, the potential risk of questioning this type of communication with the authority should not take place, though it is not possible to eliminate this risk unequivocally to a degree that guarantees certainty of effectiveness of such a solution. On the other hand, referring to beneficiaries who are public entities, the rule should be to serve documents to the electronic registry box. If so far communication has taken place in a mixed manner, or only in the traditional way, there are no reasons not to serve documents to the electronic registry box as the main form of service. In such cases the legislator assumes the electronic way as a rule, and at the same time, in the case of traditional service, it does not limit the time in which it is possible to carry out service under Article 39^2^ of the Code. Undoubtedly, the authority’s contact with public entities should involve transferring all communication in this regard to electronic circulation. Since it is possible to contact the beneficiary by means of the SL2014 application for clearing the project and for communicating with the managing authority, contact in terms of the ongoing administrative proceedings should also be possible.

## Data Availability

Data sharing not applicable.

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
