# Peer review of "Electronic Communication with Public Administration in the Time of COVID-19—Poland’s Experience"

_ijerph, 2021, doi:10.3390/ijerph18020685_

Round 1

Reviewer 1 Report

The article discusses the procedures for e-documentation under communication with the Party and Participants to proceedings from the Perspective of the Activity of the Managing Authority of the Regional Operational Program. The paper refers to the implementation of these processes in connection with the restrictions introduced on the territory of Poland (the state of epidemic threat and then the state of epidemic).

The article is of ordering, reporting and interpretation nature. The article could be supplemented with research results showing the scale and character of the phenomenon, although on the example of a selected subject. However, the article refers to information about the actual course of these processes in a fairly general manner.

Extending the work with: research results on the scale of the problem or case study or international comparisons showing existing solutions would significantly increase the number of recipients of the article and raise the attractiveness of the article for an international audience.

From the perspective of practice and current problems related to document handling in the era of the state of epidemic threat / the state of epidemic, an important part of the work is the statute of a scanned document signed by hand and sent my email. This part of the work should be considered as a significant contribution to organizing the knowledge about admissible procedures in Poland.

Some imperfections were noted in the work:

  • No explanation of the abbreviation ePUPAP, ICT
  • Incorrect translation of the public entity electronic mail box. The author further uses the "electronic registery box" (in my opinion this the right version).
  • The title is too long and therefore not attractive

Author Response

Response to Reviewer 1’s comments

With reference to the suggestion about supplementing the text with results of research demonstrating the scale and nature of the phenomenon, the author tried to obtain information from other Polish managing authorities so as to attempt to evaluate the scale of communication by electronic means, though she did not obtain statistical information, which is due to the lack of ongoing monitoring of the problem, which the author also pointed out in the text. At the same time, the author supplemented the study with information that the ongoing communication between the beneficiary and an authority proceeds be means of the “SL2014” system, i.e. the main application of the central ICT system used i.e. in the process of clearing the project and communicating with the ROP’s Managing Authority.

With reference to the comments concerning the content of the introduction, the author specified that the study concerns the situation of beneficiaries who use funding from the European Regional Development fund, the European Social Fund and the Cohesion Fund in the framework of proceedings implemented under the cohesion policy, against whom proceedings for recovering the funds have been initiated. As part of a revision of the introduction, the author focuses on the issues of communication with an authority in a situation of initiating administrative proceedings against a beneficiary of EU funds, e.g. for returning the granted funding. At the same time, the author specifies that she bases her reflections on the experience resulting from providing legal services for one of the Polish Managing Authorities of the Regional Operational Programme, which confirms the practical dimension of the manuscript.

Therefore, the following changes have been made to the text:

  1. Lines 116-124 - the change is related to the Reviewer's remark regarding the lack of reference to the literature used and its purpose is to emphasize the reason for this state of affairs,
  2. Lines 133-137 –the change is to supplement the information indicated.

With reference to the suggestion about extending the work with: research results on the scale of the problem or case study or international comparisons showing existing solutions would significantly increase the number of recipients of the article and raise the attractiveness of the article for an international audience, the author managed to reach information about implementing the same solutions for communication with a managing authority relating to evaluation, implementation and monitoring ROP’s projects, e.g. in Romania. The author also noticed, that when it comes to implementing individual solutions that facilitate the process of e.g. recovering funds and when contacting the managing authority is necessary, no detailed solutions were introduced. It is because the ePUAP is in place and in operation and electronic communication is possible, which simply got revived in the time of COVID-19, though it did not become a norm.

Therefore, the following changes have been made to the text:

  1. Lines 304-324 - the change is related to the reviewer's objection to the lack of a comparative legal analysis; the author added this excerpt to clarify the questionable points,
  2. Lines 571-573 - the change is related to the reviewer's objection to the lack of a comparative legal analysis; the author added this fragment, referring to the EU legislation in the analyzed area,
  3. Lines 620-622 – the change is to supplement the information indicated.

Referring in detail to the irregularities pointed out in the review, the author would like to say that:

  1. The abbreviation ePUAP was explained in verse 116, i.e. as “public entity's electronic mailbox”. It was changed to “public electronic registry box” as suggested by the reviewer.
  2. The abbreviation ICT was explained as part of the revision and its definition (information and communication technologies) was added.
  3. The author changed and shortened the title. The current suggested wording: Electronic communication with public administration in the time of COVID-19 - Poland’s experience

Therefore, the following changes have been made to the text:

  1. Lines 2-3 - the change is connected with the shortening of the title in accordance with the Reviewer's note,
  2. Lines 15-20 – the change is connected with shortening the title of the publication and the necessity to specify in the introduction who exactly the article concerns,
  3. Line 147 - the change is related to the Reviewer's remark regarding an error in the ePUAP definition and consists in correcting this definition,
  4. Line 249 - the change results from the need to identify three situations in which service is done by means of electronic communication,
  5. Line 254 - the change is related to the Reviewer's remark regarding an error in the definition of ICT and consists in supplementing it.

Other changes made to the article:

  1. Line 5 - The organizational structure of the Faculty of Law and Administration of the University of Szczecin changed on 1 January 2020, thus a change in affiliation is necessary
  2. Lines 33-46 - the change is related to the Reviewer 2's remark regarding the lack of discussion of the research methods used

Lines 47-54 - the change is related to the Reviewer 2's remark regarding the lack of information about the results of the analysis in the introduction

Reviewer 2 Report

In this manuscript the authors develop a very current topic perceived as relevant by both public authorities and individuals. The contemporaneity of the topic, makes it particularly challenging to design a research paper at a time when the object of study is being developed. Credit to the authors for taking a difficult path.

With this in mind, I will briefly describe the critical issues that I have identified in the manuscript and that I believe must be considered prior to its publication.

  1. The title is too long, a bit contentious and does not provide the reader with the real purpose of the study. It should be formulated in a simpler way to draw the attention of potential readers.
  2. The abstract is too general; it should instead include (briefly) a background to the topic, the critical issues raised, the purpose of the study, the method adopted and the results achieved.
  3. The literature review is not sufficient to justify the research. It is obvious that it is highly unlikely to find scientific contributions already published on such a topical subject, but it would be appropriate to find some connection, even indirect, with comparable epidemic cases already described in the literature. In case it is impossible, the authors should put even more emphasis on this lack in the body of literature.

  4. The work lacks an effective method, the authors do not indicate what methodological choices they adopt. Reading the discussion I can identify an interpretive-descriptive approach, making the research exploratory in type. This aspect needs to be strengthened.

  5. The authors formulate five research questions, however in the conclusion there is no indication of how the study answered these questions.

In conclusion, the paper is based on a fairly detailed analysis of the question, but lacks both a theoretical framework and methodological foundation. I therefore invite the authors to make the appropriate additions through a major revision before publishing their manuscript.

Author Response

Response to Reviewer 2’s comments

Referring to note No. 1:

The author changed and shortened the title. The current suggested wording: Electronic communication with public administration in the time of COVID-19 - Poland’s experience.

Therefore, the following changes have been made to the text:

  1. Lines 2-3 - the change is connected with the shortening of the title in accordance with the Reviewer's note,
  2. Lines 15-20 – the change is connected with shortening the title of the publication and the necessity to specify in the introduction who exactly the article concerns.

Referring to note No. 2:

With reference to the comments concerning the content of the introduction, the author specified that the study concerns the situation of beneficiaries who use funding from the European Regional Development fund, the European Social Fund and the Cohesion Fund in the framework of proceedings implemented under the cohesion policy, against whom proceedings for recovering the funds have been initiated. As part of a revision of the introduction, the author focuses on the issues of communication with an authority in a situation of initiating administrative proceedings against a beneficiary of EU funds, e.g. for returning the granted funding. At the same time, the author specifies that she bases her reflections on the experience resulting from providing legal services for one of the Polish Managing Authorities of the Regional Operational Programme, which confirms the practical dimension of the manuscript.

Therefore, the following changes have been made to the text:

  1. Lines 33-46 - the change is related to the Reviewer's remark regarding the lack of discussion of the research methods used,
  2. Lines 47-54 - the change is related to the Reviewer's remark regarding the lack of information about the results of the analysis in the introduction,

Referring to note No. 3:

With reference to the suggestion about supplementing the text with results of research demonstrating the scale and nature of the phenomenon, the author tried to obtain information from other Polish managing authorities so as to attempt to evaluate the scale of communication by electronic means, though she did not obtain statistical information, which is due to the lack of ongoing monitoring of the problem, which the author also pointed out in the text. At the same time, the author supplemented the study with information that the ongoing communication between the beneficiary and an authority proceeds be means of the “SL2014” system, i.e. the main application of the central ICT system used i.e. in the process of clearing the project and communicating with the ROP’s Managing Authority. The author managed to reach information about implementing the same solutions for communication with a managing authority relating to evaluation, implementation and monitoring ROP’s projects, e.g. in Romania. The author also noticed, that when it comes to implementing individual solutions that facilitate the process of e.g. recovering funds and when contacting the managing authority is necessary, no detailed solutions were introduced. It is because the ePUAP is in place and in operation and electronic communication is possible, which simply got revived in the time of COVID-19, though it did not become a norm.

In verse 108 (before revision) the author pointed out the problem of absence of relevant scholarly studies. After the review, the author emphasized these issues by demonstrating that the main source of the research material involves author’s experience resulting from providing legal services for one of the Managing Authorities in Poland, which involved requests for legal opinions addressed to the author by the authority’s employees and practical problems the author noticed as a result of practising the profession of a legal counsel. Observation of the occurring transformations and of the fossilization of legal regulations led to a number of conclusions, including a primary one about the legal provisions not being adjusted to the dynamic situation associated with COVID-19. The dynamic situation associated with the pandemic also caused an absence of scholarly studies in which authors would analyse the situation of beneficiaries of e.g. EU funds against whom proceedings for returning those funds were initiated during the pandemic.

Therefore, the following changes have been made to the text:

  1. Lines 116-124 - the change is related to the Reviewer's remark regarding the lack of reference to the literature used and its purpose is to emphasize the reason for this state of affairs,
  2. Lines 133-137 – the change is to supplement the information indicated,
  3. Lines 304-324 - the change is related to the reviewer's objection to the lack of a comparative legal analysis; the author added this excerpt to clarify the questionable points,
  4. Lines 571-573 - the change is related to the reviewer's objection to the lack of a comparative legal analysis; the author added this fragment, referring to the EU legislation in the analyzed area,
  5. Lines 620-622 – the change is to supplement the information indicated.

Referring to note No. 4:

As part of revision, the author also supplemented the text with information concerning the use of research methods (i.e. the method of interpretation of applicable laws, the analytical method and the empirical method). The author pointed out that she bases her reflections on the experience resulting from providing legal services for one of the Polish Managing Authorities of the Regional Operational Programme, using at the same time a number of research methods

(i.e. the method of interpretation of applicable laws to establish applicable provisions of the law that regulate admissibility of electronic communication with a public authority and to establish efficiency of such communication, the analytical method, applied in reference to the relevant state of the art in the achievements of legal scholarship, and the empirical method, based on observation and analysis of practical issues resulting from the author’s cooperation with a Polish managing authority).

Therefore, the following changes have been made to the text:

  1. Lines 33-46 - the change is related to the Reviewer's remark regarding the lack of discussion of the research methods used,
  2. Lines 116-124 - the change is related to the Reviewer's remark regarding the lack of reference to the literature used and its purpose is to emphasize the reason for this state of affairs.

Referring to note No. 5:

In her conclusions, the author points to the lack of introduction of comprehensive regulations (also at the EU level - for all EU Member States) in terms of de-formalizing the principles of communication in the course of pending administrative proceedings. The author notices an absence of unambiguous regulations that allow for a scanned document signed by hand and sent my email to be qualified into the category of documents served by electronic means, through use of means of electronic communication. The author assesses this absence negatively due to the fact that such action seems the simplest in a situation caused by COVID-19.

Other changes made to the article:

  1. Line 5 - The organizational structure of the Faculty of Law and Administration of the University of Szczecin changed on 1 January 2020, thus a change in affiliation is necessary,
  2. Line 147 - the change is related to the Reviewer 1's remark regarding an error in the ePUAP definition and consists in correcting this definition,
  3. Line 249 - the change results from the need to identify three situations in which service is done by means of electronic communication,
  4. Line 254 - the change is related to the Reviewer 1's remark regarding an error in the definition of ICT and consists in supplementing it.

Round 2

Reviewer 1 Report

Thank you for all the corrections made and replies.

The style of using abbreviation in the whole paper must be improve (ROB, CAP etc.).

It is difficult to find an example from Romania in the text. It is not marked appropriately.

Author Response

Dear Sir/Madam, 

Taking the reviewer's comment, I looked at the abbreviations again and thought everyone was explained, but to avoid any objection, I changed the CAP to the Code as no other code is referenced, so you I do so. I changed the insert "hereinafter referred to as ..." accordingly. Also, I inserted the full name of CAP (Code of Administrative Procedure) at the first use in each subsection (lines 158-159, 238-239, 294, 327, 376, 393, 459, 467-468, 607-608), and in subsequent uses in a given paragraph or subsection later changed to the Code (lines 211, 238-239, 294, 327, 330, 406, 422, 447, 463, 475, 477, 479, 484, 489, 492, 525, 545, 548, 556, 584, 588, 612, 629-630, 644, 653).

ROP - I deleted the abbreviation, there were two places in total and replaced with the full name of the Program (lines 129, 309-310)

ERB - as above I deleted it, entered the full name (line 472, 478, 503, 521).

I couldn't find any other abbreviations (apart from ICT, which I already explained).

Referring to the lack of recording of information on actions taken in Romania, I have completed this issue in line 305.

Reviewer 2 Report

Dear Authors, I have carefully read the latest version of your manuscript appreciating the additions and improvements you have made. Therefore, after this revision, I believe that your paper is suitable for publication in this journal.

Author Response

Dear Sir/Madam,

thank you for the review and all the critical comments that were extremely helpful and educating for me, and most of all - allowed to enrich the article I prepared. Thank you also for accepting the article and for the possibility of sending it to print

Taking the Reviewer's 1 comment, I looked at the abbreviations again and thought everyone was explained, but to avoid any objection, I changed the CAP to the Code as no other code is referenced, so you I do so. I changed the insert "hereinafter referred to as ..." accordingly. Also, I inserted the full name of CAP (Code of Administrative Procedure) at the first use in each subsection (lines 158-159, 238-239, 294, 327, 376, 393, 459, 467-468, 607-608), and in subsequent uses in a given paragraph or subsection later changed to the Code (lines 211, 238-239, 294, 327, 330, 406, 422, 447, 463, 475, 477, 479, 484, 489, 492, 525, 545, 548, 556, 584, 588, 612, 629-630, 644, 653).

ROP - I deleted the abbreviation, there were two places in total and replaced with the full name of the Program (lines 129, 309-310)

ERB - as above I deleted it, entered the full name (line 472, 478, 503, 521).

I couldn't find any other abbreviations (apart from ICT, which I already explained).

Referring to the lack of recording of information on actions taken in Romania, I have completed this issue in line 305.

Kind regards,
Aleksandra Klich 
